# Advanced Autoantibody Testing in Systemic Sclerosis

**DOI:** 10.3390/diagnostics13050851

**Published:** 2023-02-23

**Authors:** Kholoud Almaabdi, Zareen Ahmad, Sindhu R. Johnson

**Affiliations:** 1Toronto Scleroderma Program, Division of Rheumatology, Department of Medicine, Mount Sinai Hospital, University of Toronto, Toronto, ON M5T 1R8, Canada; 2Toronto Scleroderma Program, Schroeder Arthritis Institute, Division of Rheumatology, Department of Medicine, Toronto Western Hospital, Toronto, ON M5T 379, Canada; 3Institute of Health Policy Management and Evaluation, University of Toronto, Toronto, ON M5T 3M6, Canada

**Keywords:** systemic sclerosis, scleroderma, antibodies

## Abstract

Systemic sclerosis is a systemic autoimmune rheumatic disease characterized by immune abnormalities, leading to vasculopathy and fibrosis. Autoantibody testing has become an increasingly important part of diagnosis and prognostication. Clinicians have been limited to antinuclear antibody (ANA), antitopoisomerase I (also known as anti-Scl-70) antibody, and anticentromere antibody testing. Many clinicians now have improved access to an expanded profile of autoantibody testing. In this narrative review article, we review the epidemiology, clinical associations, and prognostic value of advanced autoantibody testing in people with systemic sclerosis.

## 1. Introduction

Systemic sclerosis (SSc) is a systemic autoimmune rheumatic disease (SARD) characterized by endothelial dysfunction, leading to small vessel vasculopathy, immune dysregulation, fibroblast dysfunction, and subsequent fibrosis of the skin and viscera [1,2,3,4,5]. It is a rare disease with an estimated global prevalence of 17.6 per 100,000 and an incidence rate of 1.4 per 100,000 persons per year [6]. SSc presents and evolves differently across patients, leading to progressive disability, diminished quality of life, and mortality [7].

SSc is distinguished by the presence of serum autoantibodies that target various intracellular antigens. These autoantibodies, which are present in more than 95% of patients, can be useful diagnostic indicators for SSc [8]. SSc-specific antibodies target structures in the nucleoli, nucleoplasm, or chromatin of cells that are important for cell transcription and division. The autoantibody profile of a patient with SSc does not change over time and is not affected by immunosuppressive therapy [8,9,10].

Historically, clinicians have been limited to antinuclear antibody (ANA) measured either by immunofluorescence or enzyme-linked immunosorbent assay, anti-topoisomerase (ATA, also known as anti-Scl-70) antibody, and anticentromere antibody (CENP). These autoantibodies are associated with specific clinical features and organ involvement and can inform prognosis [11]. Anti-topoisomerase I, anticentromere, and anti-RNA polymerase III antibodies were deemed vitally important in the concept of SSc [12] due to their ability to distinguish SSc from other SARD, such that they were included in the American College of Rheumatology (ACR)/European League Against Rheumatism (EULAR) classification criteria for SSc [13].

Clinicians now have access to a larger selection of SSc-specific autoantibodies. An understanding of the epidemiology of SSc-specific antibodies, clinical associations, and prognostic value will assist clinicians in their interpretation and inform the care of SSc patients.

## 2. Antinuclear Antibodies (ANA)

ANA are common in the general population, occurring in up to 20% of women. The presence of an ANA is not necessarily suggestive of a pathologic process, particularly at low titers [14]. Rather, low-titer ANA are thought to reflect a state of benign autoimmunity. However, a subset (5–8%) of these individuals will progress to develop a SARD, such as SSc, Sjogren’s syndrome, or systemic lupus erythematosus [14]. ANA-positive individuals that subsequently develop a SARD have significantly increased T and B cell activation and increased LAG3^+^ T regulatory cells and TGF-ß1 [15,16,17,18]. Immunoregulation usually prevents development of rheumatic disease in ANA-positive individuals. In contrast, immunoregulation becomes impaired in individuals who progress to develop a SARD, resulting in an imbalance favoring inflammation and fibrosis.

Since the 1960s, it has been recognized that ANA are common in individuals with SSc [19,20]. ANA have been reported to occur in 75–95% of patients with SSc, with a sensitivity of 85% and specificity of 54% on immunofluorescence [21]. The antigen substrate that is utilized for the assay affects the specificity and sensitivity of ANA differently. An indirect immunofluorescence assay using HEp-2 cells (HEp-2 IFA) is the gold standard technique. The presence of ANA as a result of HEp-2 IFA is reported as a titer and a pattern. A clinically relevant ANA titer is 1:80 or more [22].

The staining pattern reported with ANA testing by HEp-2 IFA can also be informative. The presence of anti-Scl-70 and anti-U1-RNP antibodies in the sera creates a speckled pattern, while anti-Th/To, anti-fibrillarin (anti-U3RNP) and anti-PM/Scl antibodies create a nucleolar staining pattern. Anti-RNAP I antibodies result in nucleolar staining, while antibodies against RNAP II and III give a speckled appearance or no fluorescence [21]. With the identification of over 30 staining patterns that span many diseases, an international consensus on antinuclear antibody patterns (ICAP) has proposed a classification system to standardize the interpretation and reporting of staining patterns [23] (Table 1). While the presence of ANA and staining patterns is helpful, their absence should be interpreted with caution. For example, the anti-RNAP antibodies demonstrate nucleolar staining only 30–44% of the time [24,25]. Thus, ANA staining patterns should not be used as the sole screening test for SSc-specific antibodies. ANA-negative SSc patients exist and may reflect a subset of SSc who have delayed progression of nailfold microangiopathy, defined by an early nailfold capillary NVC pattern [10].

In the following section, we describe individual autoantibodies observed in SSc, their clinical associations, and their predictive value. In Table 2, we provide an overview of the sensitivity and specificity of these antibodies in people with SSc. In Table 3, we provide an overview of their clinical associations, prognostic value, and prevalence.

### 2.1. Anti-Topoisomerase I Antibodies

Since anti-topoisomerase I antibodies (ATA) respond to immunoblots with a 70 kDa protein, they were originally known as anti-Scl-70 antibodies. Further research revealed that Scl-70 was a breakdown product of the full-length 100 kDa protein; thus, it was found that the name Scl-70 was misleading. ATA was detected in 15–42% of SSc patients, with 90–100% specificity [8,44]. ATA has sensitivity of 34% [21]. Table 2. ATA has a poor prognosis and is highly associated with diffuse cutaneous SSc (dcSSc). Patients with limited cutaneous SSc (lcSSc) and other SARD have also been noted to have ATA. The risk of severe pulmonary fibrosis and cardiac involvement is increased in SSc patients with ATA. Additionally, tendon friction rubs, the development of digital ulcers, and joint involvement have all been associated with ATA [8,29,44] (Table 3). An association with scleroderma renal crisis was reported but was not found consistently across all SSc cohorts. Furthermore, the presence of ATA in patients with Raynaud’s phenomenon is associated with a higher risk of developing SSc [29].

### 2.2. Anticentromere Antibodies

Anti-CENP antibodies, also known as anticentromere antibodies, were first reported in 1980. Several CENP proteins have been identified (CENP-A, CENP-B, CENP-C, and others), but CENP-B is thought to be the primary target of the anti-CENP B cell response in SSc [5]. Anti-CENP is the most commonly detected autoantibody in SSc cohorts, with a detection frequency of 20 to 38% [8,44,45]. Anti-CENP antibodies are specific to SSc and are reported to have specificity of 99.9% and sensitivity of 33% [26,27]. They occur less frequently in individuals of Afro-Caribbean descent compared to Caucasians [46]. Additionally, primary biliary cirrhosis, Sjogren’s syndrome, Raynaud’s phenomenon, and systemic lupus erythematosus have all been linked to anticentromere antibodies [47]. Patients with Raynaud’s phenomenon are at high risk of developing SSc if they have anti-CENP antibodies. [5,22]. When compared to other SSc-related antibodies, anti-CENP antibodies are typically associated with limited cutaneous SSc and have a better prognosis [48,49]. In this clinical subgroup of individuals, anti-CENP is associated with a higher risk of pulmonary arterial hypertension, peripheral neuropathy, and mortality [29,50]

### 2.3. Anti-Ribonucleic Acid Polymerase I, and III Antibodies

Anti-ribonucleic acid polymerase (anti-RNAP) antibodies were first described in the 1990s. Anti-RNAP I and III antibodies almost always coexist and are considered to be highly specific to SSc [8]. Anti-RNAP II antibodies are not only seen in SSc but also in systemic lupus erythematosus and overlap syndromes. Since ELISA and LIA are now more frequently used for their detection, the nucleolar speckled immunofluorescence pattern normally associated with anti-RNAP is not a sensitive tool for detecting these autoantibodies. [8,51]. The frequency of anti-RNAP I and III varies between 5% and 31% of SSc patients. In a recent meta-analysis, the pooled overall prevalence of anti-RNAP III was 11% [51]. RNAP antibodies consist of two subunits: the largest RP-155 and RP-11 [45]. RP-155 is associated with dcSSc and a higher risk of renal crisis. These patients may also be at higher risk of tendon friction rubs, synovitis, myositis, joint contractures, and the risk of developing malignancies. Despite the prevalence of renal involvement, survival is better in patients with anti-RNAP than in those with ATA or anti-U3RNP [29]. Although they have 100% specificity, autoantibodies against the RP-11 subunit of RNAP III are less sensitive than anti-RP-155 antibodies and do not seem to improve the diagnostic utility of anti-RP-155.

### 2.4. Anti-Fibrillarin (Anti-U3RNP) Antibodies

Bernstein et al. published the first report on anti-fibrillarin (anti-U3RNP) antibodies in 1982. Anti-U3-RNP antibodies specifically target a 34 kDa component of the small nucleolar ribonucleoprotein, which is located in the fibrillar area of the nucleolus and is implicated in pre-RNA processing [52]. Anti-U3RNP antibodies are detected in 41% of SSc patients. It is considered relatively specific to SSc and is mutually exclusive from CENP, ATA, and anti-RNAP [8,53]. They are found more frequently in African American than in Caucasian patients [54]. Anti-U3 RNP is associated with male sex, Afro-Caribbean descent, younger age at diagnosis, and higher risk of developing PAH and gastrointestinal involvement [55]. Regardless of demographics or disease type, anti-fibrillarin antibody positivity is associated with poorer survival. [31].

### 2.5. Anti-Th/To Ribonucleoprotein (Anti-Th/To) Antibodies

Anti-Th/To ribonucleoprotein antibodies (anti-Th/To) mainly bind to two mitochondrial RNA processing (MRP) proteins and the ribonuclease P complexes. They are present in 1–13% of SSc patients [12]. Anti-Th/To antibodies have high specificity (99%) for SSc. However, anti-Th/To antibodies have been observed in patients with rheumatoid arthritis, systemic lupus erythematosus, polymyositis, and Sjogren’s syndrome. Despite reports of up to 21% of anti-Th/To positive patients having dcSSc, the majority of these patients have lcSSc [56]. Anti-Th/To-positive SSc patients often develop pulmonary hypertension and interstitial lung disease but experience less involvement of joints and muscles [57]. Anti-Th/To antibody is a predictor for a worse prognosis [53].

### 2.6. Anti-U11/U12 RNP Antibodies

Low concentrations of macromolecular U11/U12 RNP complexes are present in eukaryotic cells, where they function as spliceosome components and catalyze the splicing of pre-messenger RNA into pre-mRNA introns [58]. The prevalence of anti-U11/U12 -RNP antibodies is 3.2% [59]. Anti-U11/U12 RNP antibodies have been associated with gastrointestinal manifestations, Raynaud’s phenomenon, pulmonary fibrosis, and an increased risk of mortality [41,60]. The presence of anti-U11/U12 RNP autoantibodies may indicate a subset of patients who are more likely to develop cancer when SSc first appears [61].

### 2.7. Anti-Ro/SSA Antibodies

Anti-SSA/Ro52 antibodies occur with a prevalence of 20% in SSc patients [62]. Anti-Ro52 antibody is a risk factor for a serious pulmonary outcome [63]. While one study found no correlation between the presence of anti-Ro52 antibodies and Raynaud’s phenomenon, sclerodactyly, digital ulcers, gangrene, calcinosis cutis, telangiectasia, or esophageal dysmotility [63], anti-Ro52 antibody is predictor of poor survival in SSc [64].

### 2.8. Anti-Ku Antibodies

Initially discovered in individuals with scleroderma–polymyositis overlap syndrome, anti-Ku antibodies were first reported in 1981 by Mimori et al. Ku is a DNA-binding protein involved in DNA repair, which is important for the non-homologous end-joining pathway’s ability to repair double-stranded DNA breaks [36]. In a recent international cohort, anti-Ku antibodies were rarely found in only 1.1% of SSc patients. Anti-Ku is more commonly detected in limited SSc patients with overlap disorders (myositis or lupus) [43,65,66]. Anti-Ku positivity is associated with myositis and interstitial lung disease (ILD), while vascular involvement is less prevalent [67]. With regard to prognosis, no survival difference has been associated with this autoantibody [67].

### 2.9. Anti-PM/Scl Antibodies

Anti-PM/Scl antibodies are a heterogeneous group of autoantibodies directed to several proteins of the nucleolar PM/Scl macromolecular complex. The two main autoantigenic protein components were identified and named PM/Scl-75 and PM/Scl-100, based on their molecular weights [68]. Anti-PM/Scl have sensitivity of 12.5% and specificity of 96.9% for SSc [69].

### 2.10. Anti-PM75 Antibodies

Anti-PM/Scl 75 antibodies occur with a prevalence of 10.4% [69]. Anti-PM75 antibodies are associated with high rates of calcinosis cutis and gastrointestinal manifestations, including gastroesophageal reflux disease, dysphagia, small intestinal bacterial overgrowth, and fecal incontinence. ILD was also prevalent in SSc patients with anti-PM75, second only to ATA-positive patients. Pulmonary hypertension is reported to be the clinical feature most commonly associated with the anti-PM75 antibody [70].

### 2.11. Anti-PM100 Antibodies

Anti-PM/Scl 100 antibodies have a prevalence of 7.1% [69]. Anti-PM100 is more associated with calcinosis rather than gastrointestinal manifestation. ILD was also less frequent compared to the anti-PM75 [70]. Patients with anti-PM100 antibodies had higher survival rates [70].

### 2.12. Anti-hUBF/NOR-90 Antibodies

The anti-NOR90 antibody, a nucleolar type of ANA, is found in 6.1% of SSc patients [71]. However, this antibody tends to be less specific for SSc and is reported in other SARD, such as systemic lupus erythematosus, Sjogren’s syndrome, and rheumatoid arthritis [39]. Anti-NOR90 antibodies may be a biomarker for idiopathic interstitial pneumonia with features of systemic sclerosis. Anti-NOR90 antibodies are associated with the occurrence of arthritis/arthralgia, sicca symptoms, and Raynaud’s phenomenon [72,73]. Systemic sclerosis with anti-NOR90 antibodies can be complicated by interstitial lung disease and cancer [40]. Anti-NOR-90 antibodies may be associated with a favorable prognosis [71].

### 2.13. Anti-RuvBL1 and RuvBL2 Antibodies

RuvBL1/2 is an important modulator of transcriptional activation and protein assembly and is essential for cell proliferation. It is located in the nucleus but can also be present in the cytoplasm [74]. Although only 1–2% of patients have anti-RuvBL1/2, it is highly specific to SSc. The relationship of anti-RuvBL1 and RuvBL2 with older onset age, more frequent diffuse skin and skeletal muscle involvement, male sex, and overlap myositis are its distinguishing features [75].

### 2.14. Platelet-Derived Growth Factor Stimulatory Antibodies (PDGFs)

Platelet-derived growth factor stimulatory (PDGF) antibodies are the primary mitogens for cells of mesenchymal and neuroectodermal origin. PDGF, first described in the 1970s as a serum factor that stimulates smooth muscle cell proliferation, is now one of the best-characterized growth factor receptor systems [76]. SSc appears to have a distinctive signature that stimulates autoantibodies against PDGFR. Their biological effect on fibroblasts may contribute to the pathogenesis of the disease. [77].

## 3. Myositis Autoantibodies

Myositis autoantibodies have traditionally been divided into subgroups of myositis-specific and myositis-associated antibodies. Myositis-specific antibodies are predominantly found in patients with polymyositis or dermatomyositis, while myositis-associated antibodies are usually found in patients with overlapping features of myositis and other SARDs [78].

### 3.1. Myositis-Specific Antibodies

Anti-synthetase antibodies target aminoacyl tRNA synthetases. The tRNA synthetases are a family of cytoplasmic enzymes that load specific amino acids onto their cognate tRNA to form an aminoacyl tRNA. There are eight anti-tRNA synthetase autoantibodies. Anti-Jo-1 antibody (directed against histidyl-tRNA synthetase) is the most common myositis-specific antibody in adults with idiopathic inflammatory myopathy, with a frequency of 9–24%. Anti-PL-12 and antibodies to the OJ, EJ, PL-7, PL-12, KS, Zo, and Ha antigens are less common [79]. The anti-SRP antibody is directed at the signal recognition particle (SRP), which participates in the translocation of newly synthesized proteins into the endoplasmic reticulum [80] The anti-MDA5 antibody is directed against RNA helicase encoded by the melanoma differentiation-associated gene 5 (MDA5). The anti-NXP-2 antibody is directed against nuclear matrix protein 2 (NXP-2) involved in transcriptional regulation. The anti-Mi-2 antibody is directed against a helicase involved in transcriptional activation. The anti-SAE antibody is directed against the small ubiquitin-like modifier activating enzyme (SAE) that regulates gene transcription.

Myositis-specific autoantibodies are only positive in 20 to 40 percent of myositis patients; therefore, a negative test does not rule out a diagnosis. Rather, a positive antibody informs the type of myositis and disease trajectory. While these antibodies have been associated with polymyositis and antisynthetase syndrome, their prevalence, clinical associations, and prognostic value in SSc is uncertain. Similarly, many antibodies have been associated with malignancy.

### 3.2. Myositis-Associated Antibodies

In contrast, myositis-associated autoantibodies are present in patients with SARD that can be associated with myositis. The presence of anti-Ro/SSA, anti-La/SSB, anti-Sm, or anti-ribonucleoprotein (RNP) antibodies in a patient with myositis suggests an association or overlap with another SARD. Anti-Ro52 antibodies are common in patients with antisynthetase antibodies, and anti-Ro60 and anti-La/SSB may be seen with other myositis-specific antibodies.

Myositis autoantibodies have rarely been described in SSc [81,82]. A recent study conducted in France showed that the prevalence of myositis-specific antibodies was 8.0% and that of myositis-associated autoantibodies was 9.7%. However, the prevalence of each antibody was low, at less than 5%. Myositis-associated autoantibodies positivity was associated with ILD and myositis, but this study showed no clinical associations with myositis-specific antibodies positivity [38,83].

## 4. Interpreting Their Operating Characteristics

It is important to remember that the sensitivity and specificity of these antibodies are affected by their prevalence and the choice of the comparator group. For example, anti-U3 RNP/fibrillarin, anti-Th/To, anti-PM/Scl, and anti-U11/U12 RNP are infrequent in the general population and other SARDs [8]. In SSc cohorts, the sensitivity and specificity of these antibodies may differ depending on several factors, such as race, area of origin, immunogenic markers, and the autoantigen immunoassay they were detected with [8,45] (Table 2). A recent study from the Netherlands compared the detection of SSc-specific autoantibodies through various diagnostic tests. Seventy-nine percent of patients tested positive for SSc autoantibodies in at least one diagnostic test. SSc-specific autoantibodies included in the American College of Rheumatology/European League Against Rheumatism (ACR/EULAR) criteria showed a high degree of concordance (antitopoisomerase, anticentromere antibody, and anti-RNA polymerase III). PM/Scl, Ku, fibrillarin, and Th-To antibodies demonstrated less concordance. A minority of patients were ATA and ARA positive [2].

## 5. Are systemic Sclerosis-Associated Antibodies Mutually Exclusive?

The SSc-specific autoantibodies are thought to be mutually exclusive [26], but there is a small body of literature on their coexistence. In a recent study of 2799 SSc patients conducted in England, 5% had more than one SSc-specific autoantibody [84]. ATA and ACA expression are not completely mutually exclusive, but their coexistence is rare (<1% of patients with SSc). Patients with both autoantibodies often have diffuse cutaneous disease and display immunogenetic features of both antibody-defined subsets of SSc [85]. In another study of 4687 patients from the EUSTAR database, 29 patients (0.6%) were documented as double positive for both ATA and CENP antibodies. Sera from 14 patients were available for central reanalysis by immunofluorescence, enzyme immunoassay, and immunoblot to confirm antibody status. Eight patients were confirmed to contain both autoantibodies. The prevalence of cutaneous and visceral manifestations in double-positive antibody patients was similar to single-positive antibody patients [86]. In an Italian cohort of 210 SSc patients, in which a commercially available LIA was used for the simultaneous detection of 13 SSc-associated autoantibodies, except for anti-Ro52/TRIM21 (specificity of 50%), all autoantibodies were very specific (from 93.3% anti-PM/Scl-75 to 100% anti-PDGFR, AFA, and anti-RP-11) for SSc. Anti-Ku was associated with another autoantibody in 0.4% of positive patients, and anti-PM/Scl-110 was associated with another autoantibody in 0.42% of patients [45].

## 6. Using Antibodies to Identify Subsets of SSc Patients

Given the association of specific organ involvement, outcomes, and antibodies, there are international initiatives to develop classification criteria to identify subsets of SSc patients [86,87,88]. The Very Early Diagnosis of Systemic Sclerosis (VEDOSS) criteria identify a subset of patients based on the presence of RP, puffy fingers, antinuclear antibodies, and capillaroscopy or SSc-specific antibodies [89]. Using the Australian Scleroderma Interest Group and the Canadian Scleroderma Research Group cohort data, three subsets of SSc were proposed [90]. Subset 1 is characterized by digital ulcers, pitting scars, and anti-topoisomerase I antibodies. Subset 2 is characterized by diffuse skin involvement, tendon friction rubs, and anti-RNA polymerase III antibodies. Subset 3 is characterized by limited or no skin involvement and anticentromere antibodies [91]. Other researchers have proposed a simplified system combining the extent of skin involvement and SSc-specific antibodies [87]. A reliable, responsive, and valid SSc subset system using an antibody profile could be used to identify patients most likely to derive a therapeutic benefit, and as a cohort enrichment strategy for trials of therapeutic agents [90,91].

## 7. Summary

Clinicians have increasing access to advanced autoantibody profiling in the assessment of patients with SSc. The staining pattern on ANA testing by indirect immunofluorescence assay on Hep-2 cells can provide a clue to the SSc-specific antibodies that may be present and that warrant further investigation. This narrative review provides up-to-date, practical information for the practicing clinician who cares for people with SSc. The presence of SSc-specific antibodies can inform the prediction of patients with Raynaud’s phenomenon who have an increased probability of developing SSc and warrant close follow-up. The presence of SSc-specific antibodies can assist with making a clinical diagnosis, can contribute to classification of SSc for research purposes, can guide monitoring for specific internal organ manifestations, and can inform prognosis.

## Figures and Tables

**Table 1 diagnostics-13-00851-t001:** Summary of antinuclear antibody staining patterns on indirect immunofluorescence, their associated antigens, and their ICAP classification.

Staining Pattern	Antigen	ICAP Classification
**Nucleolar**		
Homogeneous nucleolar	PM/Scl-75, PM/Scl-100, Th/To	AC-8
Clumpy	U3-RNP (fibrillarin)	AC-9
Punctate	RNA-Polymerase I, NOR-90	AC-10
**Speckled**		
Fine	Mi-2, TIF1γ, TIF1β, Ku	AC-4
Large/Course	U1RNP, RNA Polymerase III	AC-5
Topoisomerase I-like		AC-29
**Centromere**	CENP-A/B	AC-3

ICAP International Consensus on Antinuclear Antibody Patterns. www.ANApatterns.org accessed on 5 December 2022. AC Anti-Cell.

**Table 2 diagnostics-13-00851-t002:** Summary of antibody sensitivity and specificity in systemic sclerosis.

Antibody	Sensitivity	Specificity	Control Group	Reference
Anticentromere	33%	99.9%	Healthy controls	Reveille, J.D.,2003 [26] Spencer-Green, G., 1997 [27]
Anti-topoisomerase I	34%	99.6%	Healthy controls	Spencer-Green, G., 1997 [27]
Anti-RNA polymerase III	10–20%	98.8%	SSc	Satoh, T., et al., 2009 [28]
RP-155	5.7%	99.3%	Other SARD	Steen, V.D., 2005 [29]
RP-11	5.2%	100%	Other SARD	Steen, V.D., 2005 [29]
Anti-U3 RNP	3.74%	98.70%	SSc	Gehring, I., et al., 2017 [30]
Anti-Th/To	3.3%	98.7%	Other SARD	Steen, V.D., 2005 [29]
Anti-U11/U12	NA	100%	SSc	Fertig, N., et al., 2009 [31]
Anti-Ro52	18.1%	50%	Other SARD	Steen, V.D., 2005 [29]
Anti-Ku	4.7%	96%	Other SARD	Steen, V.D., 2005 [29]
PM/Scl-75	10.9%	93.3%	Other SARD	Steen, V.D., 2005 [29]
PM/Scl-100	6.8%	98%	Other SARD	Steen, V.D., 2005 [29]
NOR-90	4.8%	96.7%	Other SARD	Steen, V.D., 2005 [29]
Platelet-derived growth factor stimulatory antibodies	0.95%	100%	Other SARD	Steen, V.D., 2005 [29]

SSc systemic sclerosis. SARD systemic autoimmune rheumatic diseases.

**Table 3 diagnostics-13-00851-t003:** Summary of the clinical associations, prevalence, and predictive value of autoantibodies in systemic sclerosis.

Antibodies	Clinical Association	Prognosis	Prevalence	Reference
Anti-Scl-70	Severe pulmonary fibrosis and cardiac involvement, joint involvement, tendon friction rubs, digital ulcers, renal crisis, Raynaud’s phenomenon	Poor prognosis	12.3–22.5%	Choi and Fritzler 2016 [32]
			15.0%	Tangri et al., 2013 [33]
			18%	Graf et al., 2012 [34]
			15–20%	Reveille et al., 2003 [26]
Anticentromere	Raynaud’s phenomenon, pulmonary arterial hypertension (PAH), primary biliary cirrhosis	Better prognosis	42.5–67.5%	Choi and Fritzler 2016 [32]
			34.6%	Tangri et al., 2013 [33]
			40%	Graf et al., 2012 [34]
			20–30%	Reveille et al., 2003 [26]
Anti-RNA polymerase III (Anti-RP155, Anti-RP11)	Diffuse cutaneous systemic sclerosis (DcSSc), renal crisis, tendon friction rubs, synovitis, myositis, joint contractures, and risk of developing malignancy	Better prognosis	0–31.3%	Choi and Fritzler 2016 [32]
			18.5%	Tangri et al., 2013 [33]
			16%	Graf et al., 2012 [34]
			20%	Reveille et al., 2003 [26]
			6–9%	Mehra et al., 2013 [8]
Anti-fibrillarin (Anti-U3RNP)	DcSSc, renal and cardiac involvement. In African-American patients: severe pulmonary disease, pulmonary hypertension, severe small bowel involvement	Poor prognosis	4%	Reveille et al., 2003 [26]
			4–10%	Mehra et al., 2013 [8]
Anti-Ro52	Lung involvement, PAH	Poor prognosis	19.4%	Tangri et al., 2013 [33]
			34.4%	Lee et al., 2021 [35]
			21%	Meridor et al., 2021 [36]
			12%	Breda et al., 2013 [37]
Anti-NXP2	Interstitial lung disease (ILD), arthritis, dcSSc	None	0.7%	Leurs et al., 2021 [38]
Anti-hUBF/NOR-90 antibodies	Raynaud’s phenomenon, ILD, cancer	None	Few case reports	Dagher et al., 2002 [39] Yamashita et al., 2021 [40]
Anti Th/To	ILD and scleroderma renal crisis. Less joint and muscle involvement.	Poor prognosis	15%	Choi and Fritzler 2016 [32]
			6%	Graf et al., 2012 [34]
			2–5%	Reveille et al., 2003 [26]
			2–5%	Mehra et al., 2013 [8]
Anti-U11/U12 RNP	Raynaud’s phenomenon, gastrointestinal involvement, severe ILD, and higher risk of mortality	Poor prognosis	3%	Fertig et al., 2009 [41]
			3.2	Kayser et al., 2015 [42]
Anti-Ku	Myositis, arthritis, joint contractures, fingertip ulcers, telangiectasia, and ILD	Better prognosis	5%	Graf et al., 2012 [34]
			2–4%	Kayser et al., 2015 [42]
Anti-PM75	Calcinosis and gastrointestinal symptoms (gastroesophageal reflux disease [GERD], dysphagia, bacterial overgrowth, fecal incontinence), arthritis, myositis, ILD, pulmonary hypertension	Better prognosis	3.7%	Wodkowski et al., 2015 [43]
Anti-PM100	Calcinosis, arthritis, myositis, ILD, pulmonary hypertension but low rates of gastrointestinal symptoms.	Better prognosis	5.0%	Wodkowski et al., 2015 [43]
Anti-PM/Scl	Raynaud’s phenomenon, arthritis, myositis, pulmonary involvement, calcinosis, and sicca symptoms	Better prognosis	9.2%	Tangri et al., 2013 [33]
			7%	Graf et al., 2012 [34]
			3%	Reveille et al., 2003 [26]
			4–11%	Mehra et al., 2013 [8]
Myositis specific antibody (MSA)	Different clinical features	No association with prognosis in SSc ^1^	8.0%	Leurs et al., 2021 [38]
Myositis-associated autoantibody (MAA)	ILD and myositis	No association with prognosis in SSc ^1^	9.7%	Leurs et al., 2021 [38]
Anti-Jo1	Has been associated with polymyositis and antisynthetase syndrome	No association with prognosis in SSc ^1^	0.6%	Tangri et al., 2013 [33]
Anti-TIF1γ	Arthritis	No association with prognosis in SSc ^1^	4%	Leurs et al., 2021 [38]
Anti-SRP	Has been associated with myositis	No association with prognosis in SSc ^1^	0.7	Leurs et al., 2021 [38]
Anti-Mi-2 α	Calcinosis, arthritis	No association with prognosis in SSc ^1^	0	Leurs et al., 2021 [38]
Anti-Mi-2 β	Calcinosis, arthritis	No association with prognosis in SSc ^1^	0.7	Leurs et al., 2021 [38]
Anti-PL-7	ILD, arthritis	No association with prognosis in SSc ^1^	0.7	Leurs et al., 2021 [38]
Anti-PL-12	ILD	No association with prognosis in SSc ^1^	0.3	Leurs et al., 2021 [38]
Anti-MDA5	ILD	No association with prognosis in SSc ^1^	1.3	Leurs et al., 2021 [38]
Anti-EJ	ILD, calcinosis	No association with prognosis in SSc ^1^	0.3	Leurs et al., 2021 [38]
Anti-OJ	No specific clinical feature in SSc	No association with prognosis in SSc ^1^	0	Leurs et al., 2021 [38]
Anti-SAE1	Has been associated with myositis	No association with prognosis in SSc ^1^	0	Leurs et al., 2021 [38]

^1^ Note: While MSA/MAA have been associated with polymyositis or malignancy, their value in assessment of SSc patients is uncertain.

## Data Availability

Not applicable.

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
