# Peer review of "Advanced Autoantibody Testing in Systemic Sclerosis"

_diagnostics, 2023, doi:10.3390/diagnostics13050851_

Round 1
Author Response
Response: Thank you for the opportunity to revise and resubmit our manuscript. Your thoughtful and thorough review has strengthened this manuscript.
The review «advanced autoantibody testing in systemic sclerosis» by Kholoud Almaabdi and colleagues is an important contribution to the complex laboratory assessment in the case of scleroderma patients. The individual antibodies are well described. However, there is a major revision of the manuscript needed for publication.
Major concerns are: There is inconsistency in the wording of auto-antibodies / autoantibodies (Abstract), anti-nuclear /antinuclear antibodies. This should be consistent in one wording throughout the whole manuscript.
Response. We have revised it as suggested auto-antibodies and anti-nuclear antibodies.
Scl-70 is correct, Abstract: ScL70; Table 3: SL70
Response. We have revised as suggested to: anti-Scl-70.
Topoisomerase I; RNA-Polymerase III with capital “I” and not with numbers (1, 3).
Response. We have revised as suggested to: Topoisomerase I and RNA-Polymerase III throughout the manuscript.
PM/Scl-75, PM/Scl-100 (correct spelling), not PM-Scl (paragraph 2), PMscl (Table 2), PM100 / PM75 (Table 3), Pm/Scl (Table 3)
Response. We have revised as suggested to: PM/Scl-75, PM/Scl-100
Always Ro52, not RO52 (Table 3), Ro/SSA, SSA/Ro52, Ro/SS, Ro/SS-A, Ro (Table 2) (all paragraph 2.6)
Response. We have revised as suggested to: Ro52 in Tables 2, 3, and 2.6 paragraph.
Please define ILD.
Response. We have defined ILD as suggested to: interstitial lung disease.
Specific remarks:
Abstract: The phrase starting with “historically, clinicians…” should be re-phrased, as it is not complete and hard to understand.
Response. We have re-phrased the sentence as suggested to: Clinicians have been limited to anti-nuclear antibody (ANA), anti-topoisomerase I (also known as anti-ScL-70) antibody, and anti-centromere antibody testing. Many clinicians now have improved access to an expanded profile of auto-antibody testing.
Page 2, line 58: why are References [9, 10] and [11, 12] not in the same bracket: [9-12].
Response. We have revised it as suggested [9-12].
Page 2, line 76. Table 1. (also for the other tables). This is no sentence. Should be re-phrased.
Response. We have revised to: Table 1. Summary of anti-nuclear antibody staining patterns on indirect immunofluorescence, their associated antigens, and ICAP Classification.
Table 2. Summary of antibody sensitivity and specificity in systemic sclerosis.
Table 3. Summary of the clinical associations, prevalence, and predictive value of autoantibodies in systemic sclerosis.
Page 2, Just above 2.1 (line 85). Please add an introductory phrase for the following sections, that you describe the individual antibodies, and mention tables 2 and 3 as overviews.
Response. We have added: In the following section, we describe individual auto-antibodies observed in SSc, their clinical associations, and their predictive value. In Table 2 we provide an overview of the sensitivity and specificity of these antibodies in people with systemic sclerosis. In Table 3, we provide an overview of their clinical associations, prognostic value and prevalence.
Table 1: The name of AC-29 is not Topi 1, but Topoisomerase I-like.
Response. We have revised as suggested to: Topoisomerase I-like.
Table 2: please check for spelling: Topoisomerase I, RNA polymerase III, Ro, PM/Scl-
Response. We have revised as suggested to: Topoisomerase I, RNA polymerase III, Ro, PM/Scl75, and PM/Scl100.
Table 3: please check for consistency in column “antibodies”: sometimes with and sometimes without antibodies, Anti-, anti-; myositis specific (MSA) vs. Myositis-associated (MAA) autoantibodies
Response. We have revised as suggested.
Table 3: column “clinical association”: please check for punctuations (; . capital or small letter at the beginning).
Response. We have revised as suggested.
Table 3: Typo: anti-RP155
Response We have revised as suggested to: Anti-RP155.
Table 3 / anti-NXP2: Typo: Dssc – correct: dcSSc
Response. We have revised it as suggested to: dcSSc.
Table 3: why are PM/Scl-75, PM-Scl-100 and +Pm/Scl separated?
Response. We report the findings as they were reported in the literature.
Table 3, lines after myositis specific (MSA) and chapter 4/4.1: the meaning of these lines are not comprehensive, in what context are they mentioned here – anti-Jo-1, e.g., has clear clinical association, it is wrong to indicate NA in this context. Either they are not mentioned or they are e.g. mentioned as differential diagnosis in case of ILD. Furthermore, many of the MSA/MAA have clinical associations with malignancies.
Response. We have added: While MSA/MAA has been associated with polymyositis or malignancy, their value in the assessment of SSc patients is uncertain.
We have also added: Table 3: anti-Jo1 Has been associated with polymyositis and antisynthetase syndrome. Anti-SRP and anti-SAE Has been associated with myositis.
The prognosis: No association with prognosis in SSc1
In paragraph 4.1: While these antibodies have been associated with polymyositis and anti-synthetase syndrome, their prevalence, clinical association, and prognostic value in SSc is uncertain. Similarly, many antibodies have been associated with malignancy.
Paragraph 2.2: Typo: …have a better prognosis[27, 28]
Response. We have revised it as suggested to: better prognosis.
Paragraph 2.6. see above nomenclature of anti-Ro52
Response. We have revised it as suggested to anti-Ro52.
Paragraph 2.9/2.10: see above nomenclature of PM/Scl
Response. We have revised it as suggested to PM/Scl.
Paragraph 3: is it worth as a new paragraph – better as 2.13.
Response. We have revised it as suggested to 2.13.
Paragraph 4/4.1: If left in the manuscript, paragraph 4.1. should be extended. Why is the specific Synthetase only mentioned for Jo-1, and not for the others? Why are there only indications of the target antigens, and not the clinical manifestations. (see eg. Betteridge, and McHugh, PMID 26602539). Why are these lines always ending with a comma? Anti-TIF-1gamma: ??
Response. We have added: Anti-synthetase antibodies are autoantibodies that target aminoacyl tRNA synthetases. The tRNA synthetases are a family of cytoplasmic enzymes that load specific amino acids onto their cognate tRNA to form an aminoacyl tRNA. We have eight anti-tRNA synthetase autoantibodies (ASAs). Anti-Jo-1 antibody (directed against histidyl-tRNA synthetase) is the most common myositis specific antibody in adults with idiopathic inflammatory myopathy, with a frequency of 9-24%. Anti-PL-12 and antibodies to the OJ, EJ, PL-7, PL-12, KS, Zo, and Ha antigens are less common
Page 10, 2. Paragraph: “A recent study conducted in France…” The reference is missing.
Response. We have added the reference (86) Leurs, A., et al., Extended myositis-specific and associated antibodies profile in systemic sclerosis: A cross-sectional study. Joint Bone Spine, 2021. 88(1): p. 105048.
Paragraph 5: there is one interesting study comparing the results of immuno assays of different providers: Alkema et al (PMID 33818234). This study should be mentioned in this context.
Response. We have added: A recent study from the Netherlands compared the detection of SSc-specific autoantibodies by various diagnostic tests. Seventy-nine percent of patients tested positive for SSc autoantibodies in at least one diagnostic test. SSc-specific autoantibodies included in the American College of Rheumatology/European League Against Rheumatism (ACR/EULAR) criteria showed a high degree of concordance (anti-topoisomerase, anti-centromere antibody, and anti-RNA polymerase III). PM/Scl, Ku, fibrillarin and Th-To antibodies demonstrated less concordance.
Reviewer 2 Report
1) Abstract. In recent years, clinicians have had improved access to an expanded profile of auto-anti-body testing. Please, add some infromation regarding the method used to evaluate the papers for the review.
2) Abstract. In this article, we will review the epidemiology, clinical associations, and prognostic value of advanced autoantibody testing in a patient with systemic sclerosis. Please, improve this paragraph.
3) 1. Introduction L24-26 Systemic sclerosis (SSc) is a systemic autoimmune rheumatic disease (SARD) charac- terized by endothelial dysfunction leading to small vessel vasculopathy, immune dysreg- ulation, fibroblast dysfunction, and subsequent fibrosis of the skin and viscera [1]. In order to discuss the previously described points, important references are needed to be added, such as:
a- Correlation between circulating fibrocytes and dermal thickness in limited cutaneous systemic sclerosis patients: a pilot study. Rheumatol Int. 2019;39(8):1369-1376. doi:10.1007/s00296-019-04315-7
b- Serum Organ-Specific Anti-Heart and Anti-Intercalated Disk Autoantibodies as New Autoimmune Markers of Cardiac Involvement in Systemic Sclerosis: Frequency, Clinical and Prognostic Correlates. Diagnostics 2021, 11, 2165. https://doi.org/10.3390/diagnostics11112165
c- High-Resolution Computed Tomography: Lights and Shadows in Improving Care for SSc-ILD Patients. Diagnostics (Basel). 2021;11(11):1960. Published 2021 Oct 22. doi:10.3390/diagnostics11111960
4) Introduction. L 31-36. Serologically, SSc is characterized by the presence of serum autoantibodies that tar- get various intracellular antigens. These autoantibodies can be useful diagnostic indica- tors for SSc and are present in more than 95% of patients [4]. Structures that are critical for cell transcription and division that are found in the nucleoli, nucleoplasm, or chroma- tin of cells are the target of SSc-specific antibodies. The autoantibody profile of a patient with SSc does not change over the course of the patient's life and is unaffected by immu- nosuppressive therapy. Please improve this paragraph and add these references:
a- Quantification of Antifibrillarin (anti-U3 RNP) Antibodies: A New Insight for Patients with Systemic Sclerosis. Diagnostics 2021, 11, 1064. https://doi.org/10.3390/diagnostics11061064
b- Progression of nailfold microvascular damage and antinuclear antibody pattern in systemic sclerosis. J Rheumatol. 2013;40(5):634-639. doi:10.3899/jrheum.121089
5) 2. Antinuclear Antibodies (ANA) L51-53. ANA are common in the general population, occurring in up to 20% of women. The presence of an ANA is not necessarily suggestive of a pathologic process, particularly at low titres[8]. Please, underline also the link between ANA and microvascular/skin damage.
6) 8. Summary Clinicians have increasing access to advanced autoantibody profiling in the assessment of patients with SSc. The staining pattern on ANA testing by indirect immunofluorescence assay on Hep-2 cells can provide a clue to the SSc-specific antibodies that may be present and that warrant further investigation. The presence of SSc-specific antibodies can inform the prediction of the patient with Raynaud’s phenomenon who has an increased probability of developing SSc and warrant a close follow-up. The presence of SSc-specific antibodies can assist with making a clinical diagnosis, can contribute to classification of SSc for research purposes, can guide monitoring for specific internal organ manifestations, and can inform prognosis. Please, underline the novelty of the study and the clinical implications of this paper
Author Response
Thank you for the careful review of this manuscript. We hope we have revised it to your satisfaction.
Comments from Reviewer
1) Abstract. In recent years, clinicians have had improved access to an expanded profile of auto-anti-body testing. Please, add some information regarding the method used to evaluate the papers for the review.
Response. In the abstract we have added that this is a ‘narrative review.’ This article was an invited submission and not a systematic review.
2) Abstract. In this article, we will review the epidemiology, clinical associations, and prognostic value of advanced autoantibody testing in a patient with systemic sclerosis. Please, improve this paragraph.
Response. We have revised it as suggested: Clinicians have been limited to anti-nuclear antibody (ANA), anti-topoisomerase I (also known as anti-ScL-70) antibody, and anti-centromere antibody testing . Many clinicians now have improved access to an expanded profile of auto-antibody testing. In this narrative review article, we will review the epidemiology, clinical associations, and prognostic value of advanced auto-antibody testing in people with systemic sclerosis.
3) 1. Introduction L24-26 Systemic sclerosis (SSc) is a systemic autoimmune rheumatic disease (SARD) characterized by endothelial dysfunction leading to small vessel vasculopathy, immune dysregulation, fibroblast dysfunction, and subsequent fibrosis of the skin and viscera [1]. In order to discuss the previously described points, important references are needed to be added, such as:
a- Correlation between circulating fibrocytes and dermal thickness in limited cutaneous systemic sclerosis patients: a pilot study. Rheumatol Int. 2019;39(8):1369-1376. doi:10.1007/s00296-019-04315-7
Response: This reference was added.
b- Serum Organ-Specific Anti-Heart and Anti-Intercalated Disk Autoantibodies as New Autoimmune Markers of Cardiac Involvement in Systemic Sclerosis: Frequency, Clinical and Prognostic Correlates. Diagnostics 2021, 11, 2165. https://doi.org/10.3390/diagnostics11112165
Response: This reference was added.
c- High-Resolution Computed Tomography: Lights and Shadows in Improving Care for SSc-ILD Patients. Diagnostics (Basel). 2021;11(11):1960. Published 2021 Oct 22. doi:10.3390/diagnostics11111960
Response: This reference was added.
4) Introduction. L 31-36. Serologically, SSc is characterized by the presence of serum autoantibodies that target various intracellular antigens. These autoantibodies can be useful diagnostic indicators for SSc and are present in more than 95% of patients [4]. Structures that are critical for cell transcription and division that are found in the nucleoli, nucleoplasm, or chromatin of cells are the target of SSc-specific antibodies. The autoantibody profile of a patient with SSc does not change over the course of the patient's life and is unaffected by immunosuppressive therapy. Please improve this paragraph and add these references:
Response. We have re-phrase it to: SSc-specific antibodies target structures in the nucleoli, nucleoplasm, or chromatin of cells that are important for cell transcription and division. The auto-antibody profile of a patient with SSc does not change over time and is not affected by immunosuppressive therapy. SSc-specific antibodies target structures in the nucleoli, nucleoplasm, or chromatin of cells that are important for cell transcription and division. The auto-antibody profile of a patient with SSc does not change over time and is not affected by immunosuppressive therapy.
a- Quantification of Antifibrillarin (anti-U3 RNP) Antibodies: A New Insight for Patients with Systemic Sclerosis. Diagnostics 2021, 11, 1064. https://doi.org/10.3390/diagnostics11061064
Response: This reference was added.
b- Progression of nailfold microvascular damage and antinuclear antibody pattern in systemic sclerosis. J Rheumatol. 2013;40(5):634-639. doi:10.3899/jrheum.121089
Response: This reference was added.
5) 2. Antinuclear Antibodies (ANA) L51-53. ANA are common in the general population, occurring in up to 20% of women. The presence of an ANA is not necessarily suggestive of a pathologic process, particularly at low titres[8]. Please, underline also the link between ANA and microvascular/skin damage.
Response. We have added: ANA-negative SSc patients exist and may reflect a subset of SSc who have delayed progression of nailfold microangiopathy, defined by an early nailfold capillary NVC pattern. This reference was added.
6) 8. Summary Clinicians have increasing access to advanced autoantibody profiling in the assessment of patients with SSc. The staining pattern on ANA testing by indirect immunofluorescence assay on Hep-2 cells can provide a clue to the SSc-specific antibodies that may be present and that warrant further investigation. The presence of SSc-specific antibodies can inform the prediction of the patient with Raynaud’s phenomenon who has an increased probability of developing SSc and warrant a close follow-up. The presence of SSc-specific antibodies can assist with making a clinical diagnosis, can contribute to classification of SSc for research purposes, can guide monitoring for specific internal organ manifestations, and can inform prognosis. Please, underline the novelty of the study and the clinical implications of this paper
Response. We have revised it as suggested to: Clinicians have increasing access to advanced auto-antibody profiling in the assessment of patients with SSc. The staining pattern on ANA testing by indirect immunofluorescence assay on Hep-2 cells can provide a clue to the SSc-specific antibodies that may be present and that warrant further investigation. This narrative review synthesis up to date yet pragmatic information for the practicing clinician who cares for people with SSc. The presence of SSc-specific antibodies can inform the prediction of the patient with Raynaud’s phenomenon who has an increased probability of developing SSc and warrant a close follow-up. The presence of SSc-specific antibodies can assist with making a clinical diagnosis, can contribute to the classification of SSc for research purposes, can guide monitoring for specific internal organ manifestations, and can inform prognosis.
Round 2
Reviewer 1 Report
all comments are well replied, the quality of the manuscript improved.
some small typos are still in.
- Table 1: large course speckled is wrong, the pattern is called large/coarse speckled
heading 2.13. not bold but italics.
Author Response
Thank you for taking the time to review our manuscript again.
- all comments are well replied, the quality of the manuscript improved. RESPONSE: Thank you.
2. some small typos are still in.
Table 1: large course speckled is wrong, the pattern is called large/coarse speckled
RESPONSE: We have revised as suggested.
3. heading 2.13. not bold but italics.
RESPONSE: We have revised as suggested.
Thank you again for your careful review. We hope the manuscript meets with your approval.